# Chemokine Dysregulation and Neuroinflammation in Schizophrenia: A Systematic Review

**DOI:** 10.3390/ijms24032215

**Published:** 2023-01-22

**Authors:** Evgeny A. Ermakov, Irina A. Mednova, Anastasiia S. Boiko, Valentina N. Buneva, Svetlana A. Ivanova

**Affiliations:** 1Institute of Chemical Biology and Fundamental Medicine, Siberian Branch of the Russian Academy of Sciences, 630090 Novosibirsk, Russia; 2Mental Health Research Institute, Tomsk National Research Medical Center of the Russian Academy of Sciences, 634014 Tomsk, Russia

**Keywords:** schizophrenia, chemokines, cytokines, receptor, inflammation, neuroinflammation, CXCL8, IL-8, CCL2, CCL4

## Abstract

Chemokines are known to be immunoregulatory proteins involved not only in lymphocyte chemotaxis to the site of inflammation, but also in neuromodulation, neurogenesis, and neurotransmission. Multiple lines of evidence suggest a peripheral proinflammatory state and neuroinflammation in at least a third of patients with schizophrenia. Therefore, chemokines can be active players in these processes. In this systematic review, we analyzed the available data on chemokine dysregulation in schizophrenia and the association of chemokines with neuroinflammation. It has been shown that there is a genetic association of chemokine and chemokine receptor gene polymorphisms in schizophrenia. Besides, the most reliable data confirmed by the results of meta-analyses showed an increase in CXCL8/IL-8, CCL2/MCP-1, CCL4/MIP-1β, CCL11/eotaxin-1 in the blood of patients with schizophrenia. An increase in CXCL8 has been found in cerebrospinal fluid, but other chemokines have been less well studied. Increased/decreased expression of genes of chemokine and their receptors have been found in different areas of the brain and peripheral immune cells. The peripheral proinflammatory state may influence the expression of chemokines since their expression is regulated by pro- and anti-inflammatory cytokines. Mouse models have shown an association of schizophrenia with dysregulation of the CX3CL1-CX3CR1 and CXCL12-CXCR4 axes. Altogether, dysregulation in chemokine expression may contribute to neuroinflammation in schizophrenia. In conclusion, this evidence indicates the involvement of chemokines in the neurobiological processes associated with schizophrenia.

## 1. Introduction

Cytokines and chemokines are signaling proteins that integrally regulate the proliferation and activation of immune cells [1]. Cytokine and chemokine dysregulation are associated with many diseases, including schizophrenia (SZ) [2]. However, chemokines have traditionally received less attention than pro- and anti-inflammatory cytokines in SZ. Chemokines are known to be a portent attractant of immune cells to the site of inflammation [3]. However, chemokines may be involved in neuromodulation, neurotransmission, and neurogenesis beyond their classical chemotactic functions [3]. Therefore, chemokines may be involved in many neurobiological processes potentially associated with mental disorders.

Chemokines are divided into four subfamilies based on the location and distance between cysteines in the amino acid sequence: CC (C-C motif chemokine ligands—CCL), CXC (C-X-C motif chemokine ligands—CXCL), CX3C (C-X3-C motif chemokine ligands—CX3CL) and C (C motif ligands—XCL) chemokines [4]. Chemokine receptors, which are also divided into four subtypes based on the subfamily of activating chemokine ligands, have been found on the surface of mainly immune cells [5]. Chemokine dysregulation is associated with the pathogenesis of autoimmune and neurological diseases [6,7]. Neuroinflammatory diseases are accompanied by an increase in blood chemokines [7]. Chemokines have a pleiotropic effect and exacerbate inflammation. SZ is known to be associated with a chronic proinflammatory state [2]. Therefore, chemokines may play an important role in neuroinflammation in SZ.

The aim of this systematic review is to summarize the most valuable evidence on chemokine dysregulation in SZ and the association of chemokines with neuroinflammation.

## 2. Methods

We adhered to the PRISMA 2020 guidelines [8,9]. PRISMA 2020 flowchart diagram is presented in Figure 1 [10]. We used the following databases for literature searches: Web of Science, Scopus, PubMed, and Google Scholar. The required information was retrieved from database inception to November 2022. The main search terms were “psychosis”, “psychotic”, “schizophrenia” and “schizoaffective”. One of these terms was used in combination with one or more of the following search terms: “chemokine”, “chemokine receptor”, “polymorphisms”, “gene polymorphisms”, “SNP”, “level”, “blood”, “CSF”, “IL-8”, “neuroinflammation”, “animal model”, “MIA model”, “22q11”and “meta-analysis”. Bibliographic lists of articles were reviewed for relevant reports. One researcher analyzed each request and assessed the relevance of the studies. The first 50 results automatically ranked by relevance of each query were analyzed. Automation tools were not used to remove records. The final reference list was formed after the consensus of all authors regarding the relevance of the reports. The main results of this systematic review are summarized in the tables. A total of 59 studies were included in the systematic review. But the bibliographic list contains more sources to expand the context of the review.

## 3. Results and Discussion

### 3.1. Association of Chemokine and Chemokine Receptor Gene Polymorphisms with SZ

Polymorphisms of genes encoding chemokines and their receptors can affect their functional activity. Available data on the genetic association of polymorphic variants of chemokine genes and their receptors with SZ are presented in Table 1.

Genetic polymorphisms in *CCL2* gene have been well studied in SZ. Saoud et al. (2019) showed that three functional polymorphisms (rs1024611, rs285765, and rs3917887), which positively regulate the expression of the CCL2 chemokine also known as monocyte chemoattractant protein 1 (MCP-1), could have a protective effect on susceptibility to SZ [12]. rs1024611, also known as the -2518A>G SNP because of its position in the promoter region of the monocyte chemoattractant protein-1 (MCP-1) as the *CCL2* gene is also called, affects the production of the corresponding protein. In this study, the higher prevalence of the MCP-1 -2518G allele in controls compared to patients with SZ suggests that the minor G allele acts as a protective factor against SZ [12].

Contradicting these findings were the results of Zakharyan et al. (2012) on the association of the -2518A>G genetic polymorphism and elevated plasma levels of MCP-1 with SZ because they found an association of the -2518G minor allele with a higher prevalence of SZ [17]. Possible explanations for these opposite results are ethnic differences in the frequency of the alleles, relatively small group size for genetic research and studying heterogeneous nosological forms of SZ. Saoud et al. presented the results of research on 200 patients and 200 healthy subjects in the Tunisian population; 45.9% of patients met the diagnostic criteria for undifferentiated, 32% for paranoid and 19.8% for disorganized subtypes of SZ [12]. The article by Zakharyan et al. (2012) studied 103 neuroleptic-treated chronic patients with paranoid SZ and 105 healthy subjects from the Armenian population [17].

Apart from these two studies, the results of several other studies indicate the non-existence of a significant association between any allele and genotype of *CCL2* and SZ [14,15,16]. The distributions of genotypes and alleles of rs1024611 polymorphic variants in patients with SZ were not significantly different from those of control individuals from the Korean population [14]. However, these distributions were marginally different when the presence of positive and negative symptomatology was taken into account. In the Italian population, no significant genotypic or allelic association was found between the A-2518G variant and SZ and no differences in age of onset [15]. However, a significant genotypic association was found between the A-2518G variant and resistance to antipsychotic treatment, with resistant patients more often carrying the G allele [16]. The latter is consistent with the findings of Xiong et al. (2014) who found that four SNPs (rs4795893, rs1024611, rs4586 and rs2857657) were significantly associated with risperidone treatment [13]. These results could indicate that there may be some effect of variations in the *CCL2* gene on the therapeutic efficacy of risperidone, and that the associated polymorphisms may be a potential genetic marker for predicting the therapeutic effect of risperidone. Dasdemir et al. (2016) analyzed the role of MCP-1-A2518G, SDF-1-3′A, CCR5-delta32, CCR5-A55029G, CXCR4-C138T and CCR2-G190A gene polymorphisms in the Turkish population to demonstrate the relationship between variants of these chemokine genes and the risk of SZ [16]. A point mutation is known as CCR2-V64I (rs1799864), in which guanine-to-adenine substitution leads to replacing amino acid valine (V) on isoleucine (I). The frequency of the A genotype of CCR5-A55029G and the AG genotype of CCR5-A55029G were found to be higher in patients than in controls and also the wild-type of CCR2-G190A, the A haplotype of CCR5-A55029G, the 190A haplotype of CCR2-G190A and the A haplotype of CCR5-A55029G were still significantly associated after Bonferroni correction. However, no significant association was found for any of the other polymorphisms with the likelihood of having SZ. Their findings suggest that the aforementioned polymorphisms may well be associated with the pathogenesis of SZ.

The relationship between SZ and the genetic polymorphisms in other chemokines has been sparsely studied. CX3C motif chemokine receptor 1 (CX3CR1), also known as the fractalkine receptor or G-protein coupled receptor 13 (GPR13), is a transmembrane protein of the G-protein coupled receptor 1 (GPCR1) family and is encoded by the *CX3CR1* gene. Polymorphisms of *CX3CR1* have been associated with diseases of the nervous system, such as Alzheimer’s disease. Three missense variants in the *CX3CR1* gene (414G>T leads to amino acids replacement Met138Ile (rs758302878); 335G>C leads to Gly112Ala (rs201442030), 163G>A leads to Ala55Thr (rs750585901) were investigated in a large Japanese cohort (the case-control sample set consisted of 2283 people with SZ and 3827 control subjects; the targeted-resequencing discovery cohort consisted of 370 people with SZ) and a significant association was found between the 163G>A in *CX3CR1* with having SZ [19]. Results of searching for an association between regions of the *CCR3* gene (intron 1; rs9853223, rs6441948, rs13326331, and rs7652290, intron 2; rs1491962) with the risk of SZ indicate that *CCR3* does not contribute to genetic susceptibility to SZ [20].

Saoud et al. (2022) investigated the impact of the polymorphisms rs2107538, rs2280788 and rs2280789 in the *CCL5* gene, as well as rs333 within the *CCR5* gene and the development of SZ in a cohort of Tunisian individuals [11]. Of the polymorphisms analyzed, only rs2107538 was associated with reduced occurrence of SZ and that more specifically in male individuals. This protective effect for all subtypes of SZ remained valid for the undifferentiated form. Moreover, this SNP affected the symptomatology of the patients. When focusing on haplotypes, they noted that the genetic combination ACT of rs2107538-rs2280788-rs2280789, with only one mutated allele rs2107538A, showed a reduced frequency in both individuals with SZ (as a whole) and those with SZ of the undifferentiated subtype [11]. The haplotype distribution profile implies that the A allele on rs2107538 could induce a protective effect by increasing production of the chemokine Regulated upon Activation Normal T-cell Expressed and Secreted (RANTES) which is the product encoded by *CCL5* gene [11].

Age-related elevations of CCL11 are associated with cognitive impairments in executive functions and episodic and semantic memory, and therefore this chemokine has been referred to as an “Endogenous Cognition Deteriorating Chemokine” or an “Accelerated Brain Aging Chemokine.” In SZ, however, elevated CCL11 levels are associated not only with impairments in cognitive functions, but also with important symptoms such as formal thought disorders [22]. Four SNPs in promoter region of the *CCL11* gene [rs17809012 (-384T>C), rs16969415 (-426C>T), rs17735961 (-488C>A) and rs4795896 (576G>A)] encoding this protein (also known as eosinophil chemotactic protein and eotaxin-1) were genotyped by Kang et al. (2018) [18]. The frequency of *CCL11* rs4795896 (-576G>A) showed a significant association with SZ in a recessive model and in a log-additive model. The allele frequency of rs4795896 also showed a significant association with SZ. Furthermore, haplotype analysis showed that GCT, ACT and GCC haplotypes of rs4795896-rs17735961-rs17809012 were significantly associated with SZ [18].

The *CXCL8* gene encodes for a protein commonly called interleukin-8 (IL-8), a member of the CXC chemokine family and an important mediator of the inflammatory response. Of the polymorphisms analyzed in the *CXCL8* gene (rs4073, rs2227306 and rs1126647), only rs1126647 showed a significant association with the risk of SZ [21].

Thus, the literature data indicate the presence of associations between the risk of SZ and polymorphic variants of genes encoding chemokines and their receptors. These associations also involve leading clinical symptoms or response to treatment with antipsychotics. This complements and extends findings regarding immune dysfunction in this disorder. Further research is needed to determine the precise effect of these polymorphisms and to find potential protective or predisposing alleles that can be used as pharmacogenetic biomarkers of SZ risk.

### 3.2. Blood Chemokine Levels in SZ

Changes in blood cytokine and chemokine levels may be associated with the neurobiological processes leading to SZ. Cytokine dysregulation in SZ has been confirmed by many studies [2]. There is much less research on chemokines in SZ. Meta-analyses are considered a more reliable source of evidence than case-control and cohort studies. Therefore, we systematized meta-analytic data on blood levels of chemokines in patients with SZ and healthy controls (Table 2). Data from narrative review by Dawidowski et al. (2021) [23] and a meta-analysis by Frydecka et al. (2018) [24] were used for systematization.

Most of the meta-analytical data indicate the dysregulation of CXCL8 (IL-8) in the blood of patients with SZ [25,26,27,28,29,30]. In the first episode psychosis (FEP) patients, a meta-analysis by Çakici et al. (2020), which included the largest number of studies compared to other meta-analyses on this topic, showed an increase in CXCL8 blood levels in SZ (Table 2) [27]. Three meta-analyses indicate an increase in CXCL8 in acute relapsed chronic (ARCh) patients [24,25,28]. In contrast, two meta-analyses found no significant differences in CXCL8 levels in familial high-risk (FHR), clinical high-risk (CHR), and ultra-high-risk (UHR) of psychosis individuals [29,30]. Interestingly, a meta-analysis by Romeo et al. (2018) showed that pharmacotherapy does not affect CXCL8 levels [31]. Thus, the available data indicate an increase in CXCL8 blood levels in FEP and ARCh patients, but no difference in FHR, CHR and UHR people.

A meta-analysis by Frydecka et al. (2018) presented compelling evidence for an increase in CCL2 (MCP-1) blood levels in SZ [24]. In a meta-analysis by Çakici et al. (2020) found no difference in FEP patients as fewer studies were analyzed [27]. Thus, the meta-analytical data support high levels of CCL2 levels in FEP and ARCh patients [24].

There are fewer studies on other chemokine levels in SZ (Table 2). Meta-analysis results by Frydecka et al. (2018) indicate an increase in CCL4 (or macrophage inflammatory protein 1β, MIP-1β) and CCL11 (eotaxin 1) in a pooled group (FEP and ARCh) of patients [24]. Subgroup analysis revealed an increase in CCL11 in ARCh patients. CCL4 blood level did not change in ARCh patients. Blood levels of CCL3 (MIP-1α), CX3CL1 (fractalkine), and CXCL10 (IP-10) did not change significantly [24].

Altogether, the evidence presented indicates an increase in CXCL8, CCL4, CCL2, and CCL11 in SZ. These chemokines may be pathophysiologically relevant biomarkers or therapeutic targets in SZ. An increase in these chemokines in the blood may be associated with the etiopathogenesis of SZ. For example, an increase in CCL4 is thought to contribute to blood-brain barrier (BBB) dysfunction [32]. High levels of chemokines in SZ may also reflect a general proinflammatory state [23], since proinflammatory cytokines stimulate the formation of chemokines (for details, see Section 3.7). However, only a few chemokines have been investigated in SZ out of at least 46 known chemokines. Further research should focus on the analysis of unstudied chemokines in SZ.

### 3.3. Chemokines in Cerebrospinal Fluid in SZ

Neuroinflammation is reflected in abnormalities of the cerebrospinal fluid (CSF). CSF is the biological fluid closest to the brain that can be used to assess inflammation parameters by lumbar puncture. However, this procedure is routine for diagnosing neurological diseases, but not psychiatric disorders. Therefore, there are few studies of inflammatory markers in the CSF of schizophrenic patients due to the inaccessibility of the material for analysis. Nevertheless, CSF studies have revealed an elevated CSF/serum albumin ratio in patients with SZ [33,34,35,36] indicating increased BBB permeability. Besides, other studies have shown an increase in CSF IgG index and the presence of oligoclonal bands in patients with SZ [34,37]. Moreover, there are data on elevated cell counts and signs of lymphocyte activation in CSF of patients [37,38,39]. Additionally, three meta-analyses found an increase in CSF levels of IL-6 and other proinflammatory cytokines in SZ [40,41,42]. This evidence indicates increased permeability of BBB to immune cell, neuroinflammation and intrathecal immunoglobulin production in the CSF of subgroup of SZ patients. Chemokines facilitate the entry of immune cells into the CNS. Available data on chemokine levels in CSF are presented in Table 3.

CXCL8 is a well-studied CSF chemokine in schizophrenic patients. Three meta-analyses have shown an increase in CSF concentrations of CXCL8 in SZ [40,41,42]. Scheiber et al. (2022) showed that CXCL8 levels were higher in CSF than in plasma in treatment-resistant SZ patients [45]. Other chemokines are much less well studied. Nikkila et al. (2002) found no significant differences in CSF concentrations of CCL3 (MIP-1α) [43]. However, this study has a small sample size (8 SZ vs. 8 HS) and low statistical power. In a recent work, Malmqvist et al. (2019) analyzed the following chemokines in CSF: CCL3, CCL4, CCL11, CCL13, CCL17, CCL22, CCL26, CXCL8 and CXCL10 [44]. However, only CXCL10 (IP-10) was detected. CSF levels of CXCL10 did not differ significantly between patients and controls or between drug-naïve patients and patients medicated with antipsychotics. Other chemokines had <50% reliable detectable values [44].

Thus, to date, only an increase in CSF levels of CXCL8 has been confirmed in SZ. It is possible that future studies will be able to evaluate the CSF concentrations of other chemokines.

### 3.4. Chemokine Expression in the Brain in SZ

Chemokines are involved in signaling between neurons and microglia and can facilitate migration and adhesion of peripheral immune cells [46]. However, the number of studies describing the expression of chemokines in the brain of patients with SZ is limited in the literature due to the complexity obtaining biological material. We summarized data from 14 studies of chemokine expression in the post-mortem brains of SZ patients (Table 4).

The most studied chemokine was CXCL8 also known as IL-8. CXCL8 is a neutrophil attractant. No differences were found in the mRNA *CXCL8* in the substance nigra [51] and Brodman’s Area (BA) 9 of prefrontal cortex [49,50], despite a statistically significant decrease in CXCL8 protein expression in the prefrontal cortex of patients SZ [50]. A significant decrease in the mRNA level of *CXCL8* in patients with SZ compared with controls was registered in dorsolateral prefrontal cortex [47] and BA 22 of left superior temporal cortex [56]. *CXCL8* mRNA extracted from cortical gray matter from the middle frontal gyrus, showed a significant diagnostic effect with a decreased expression in patients with SZ compared with healthy controls and bipolar disorder [48]. However, *CXCL8* mRNA expression did not change significantly in the midbrain of the patients [53]. Childers et al. divided patients with SZ into two subgroups based on the expression of various genes in the dorsolateral prefrontal cortex and examined the biological differences between the subgroups [54]. The authors showed an increase in mRNA expression levels of *CXCL8*, *CXCL1*, *CXCL2* and *CCL2* in Type 2 SZ subgroup (with predominant expression of neuroinflammation genes) compared to controls and the other SZ subtype [54]. These data confirm the heterogeneity of SZ and the existence of subgroups of patients with increased expression of neuroinflammation genes in the brain. However, Weickert C.S. with colleagues were among the first to identify the presence of a subgroup of patients with high inflammation [47,48]. In a recent work, a team of these authors showed that *CCL2* mRNA expression was increased in the total group of patients with SZ and in the subgroup with high inflammation compared with the control in the dorsolateral prefrontal cortex [52]. On the contrary, the same work showed a decrease in *CXCL8* mRNA in SZ [52]. The work of other authors revealed a decrease in CCL2 and CCL8 expression in the BA 22 of left superior temporal cortex of patients with SZ [56]. Thus, most data indicate a decrease in *CXCL8* and an increase in *CCL2* expression in the dorsolateral prefrontal cortex of patients with SZ. CCL2 and CXCL8/IL-8 are known to increase blood-brain barrier permeability [61,62] resulting in the recruitment of monocyte, macrophages and activated T cells to damaged CNS tissue [63]. Therefore, increased expression of *CCL2* can promote the migration of leukocytes into the brain tissue and activate neuroinflammation. Interestingly, *CCL2* expression is upregulated by IL-1β, transforming growth factor-β (TGFβ), and tumor necrosis factor-α (TNFα). Meta-analyses indicate an increase in peripheral levels of IL-1β, TGFβ and TNFα in various stages of SZ [25,28]. Besides, activated microglia are known to produce IL-1β, IL-6 and TNFα [64]. There is some evidence of increased expression of *IL-1β* and *TNFα* in the postmortem brain of patients with SZ [65]. Therefore, the increase in *CCL2* expression in the brain may be associated with an increase in both peripheral and brain expression of *IL-1β*, *TNFα* and other cytokines.

There are few data on brain expression of other chemokines in SZ. *CCL3* mRNA expression was significantly reduced in BA 46 of the dorsolateral prefrontal cortex [55] and BA 22 of left superior temporal cortex [56]. *CXCL12* mRNA was decreased in the caudate nucleus and the subependymal zone in SZ compared to healthy individuals [58]. CXCL12 expression positively correlated with markers of neural stem cells and neuronal progenitors [58]. Additionally, CXCL12 expression decreased in the microdissected neuronal layer of the olfactory epithelium from SZ patients compared to controls [59]. *CX3CL1* mRNA was unchanged in the dorsolateral prefrontal cortex patients with SZ, but protein levels of CX3CL1 were significantly lower in SZ relative to controls [57].

Gandal et al. conducted a large-scale and complex meta-analysis of microarray data on gene expression in the cerebral cortex of patients with SZ and other neuropsychiatric disorders [60]. The mRNA expression of *CCL3*, *CCL4*, *CXCL12*, *CXCL14* and *CX3CL1* was found to be downregulated in the cortex of SZ patients. However, after adjusting for multiple comparisons, only *CCL3*, *CXCL12*, and *CX3CL1* remained significant. Expression of other chemokines such as *CCL1*, *CCL2*, *CCL5*, *CCL7*, *CCL8*, *CCL11*, *CCL13*, *CCL14*, *CCL16*, *CCL18*, *CCL19*, *CCL1*, *CCL20*, *CCL21*, *CCL22*, *CCL24*, *CCL25*, *CXCL1*, *CXCL2*, *CXCL3*, *CXCL5*, *CXCL6*, *CXCL9*, *CXCL10*, *CXCL11*, *CXCL13*, *XCL1* and *XCL2* did not change significantly. Downregulation of *CXCL12* and *CXCL14* was validated using RNA-seq data sets [60]. Moreover, the analysis of transcriptomic data revealed anomalies in expression level, local splicing and transcript isoform expression of immune genes, including chemokine genes [66]. Thus, in the cerebral cortex of patients with SZ, not only anomalies in the expression of chemokine genes, but also isoforms and splicing variants are observed.

Overall, these data indicate dysregulation of chemokine expression in various areas of the brain of patients with SZ, although the results are conflicting. Increased expression of chemokines in the brain, on the one hand, can increase the recruitment of immune cells to the brain; on the other hand, this may be associated with elevated levels of peripheral cytokines. However, there are insufficient data on expression of chemokines in the central nervous system in SZ. Most of these studies have focused on the dorsolateral prefrontal cortex of patients. Further research is needed to elucidate characteristic chemokine expression patterns in other areas of the brain.

### 3.5. Expression of Chemokine Receptors in SZ

Chemokine receptors expressed on various immune cells are involved in the realization of the biological effects of chemokines. We summarized the available data on chemokine receptor expression in SZ in Table 5.

Most studies of chemokine receptor expression have been performed using brain tissue. The mRNA levels of *CCR1* in BA 22 of left superior temporal cortex in patients with SZ were lower than controls [56]. Protein and mRNA levels of *CX3CR1* in the dorsolateral prefrontal cortex and anterior cingulate cortex did not differ between SZ patients and healthy controls [57,68]. Hill et al. (2021) demonstrated a significant difference in the strength of association between CX3CR1 and pre-synaptic protein SNAP-25 in patients with SZ relative to controls [57]. *CX3CR1* mRNA expression was increased in anterior cingulate cortex of patients with SZ without suicide as compared to suicide completers [68]. *CX3CR1* mRNA in the BA 21 of temporal cortex and hippocampus were significantly decreased in SZ patients compared with healthy individuals [69,70]. Thus, CX3CR1 expression levels vary by region of the brain.

Expression of CXCR4 and CXCR7 also showed differences depending on the brain region. Volk et al. (2015) showed that elevated mRNA levels for *CXCR7* and *CXCR4* were found in BA 9 of the prefrontal cortex in SZ subjects compared with healthy individuals [67]. *CXCR7* mRNA levels were inversely correlated with mRNA levels for GABA synthesizing enzyme—GAD67, parvalbumin, somatostatin and transcription factor Lhx6 in SZ but not in healthy subjects. An increase in CXCR7 and CXCR4 may represent a compensatory mechanism to maintain migration and correct positioning of cortical parvalbumin and somatostatin neurons in SZ [67]. In the subependymal zone, *CXCR4* and *CXCR7* mRNAs were decreased in SZ compared to controls [58]. In the caudate nucleus, *CXCR4* mRNA was significantly lower in SZ compared to healthy group, while *CXCR7* expression did not significantly differ across groups [58]. The authors observed a positive correlation between *CXCR4* and *CXCR7* expression with markers of neural stem cells and neuronal progenitors. *CXCR4* and *CXCR7* also were associated with markers of inhibitory interneurons: *CXCR4* expression positively correlated with CALB2 mRNA, while *CXCR7* expression negatively correlated with neuropeptide Y mRNA [58]. Lanz et al. (2019) demonstrated that *CXCR4* expression was unchanged, but downstream signaling genes and *CXCL12* (CXCR4 ligand) were downregulated in the prefrontal cortex and striatum in SZ [71]. Another study showed that *CXCR4* expression in the neuronal layer of the olfactory epithelium did not change [59].

All of the above studies were performed on relatively small samples. Gandal et al. conducted a meta-analysis of microarray and RNAsec data on a large sample (159 patients and 293 healthy subjects) [60]. *CCR8* and *XCR1* expression have been shown to be increased in the cerebral cortex of patients with SZ based on microarray data. RNAsec data indicated a decrease in *CX3CR1* expression [60]. The expression of other chemokine receptors such as *CCR1*, *CCR2*, *CCR3*, *CCR4*, *CCR5*, *CCR7*, *CCR9*, *CCR10*, *CCRL2*, *CXCR1*, *CXCR2*, *CXCR3*, *CXCR4*, *CXCR5* and *CXCR6* did not change significantly. Besides, abnormalities in *CX3CR1* expression of isoforms and splicing variants have been found in the cortex of patients with SZ [66].

There are fewer studies of chemokine receptor expression in tissues other than the brain (Table 2). *CX3CR1* expression was studied in peripheral blood mononuclear cells (PBMC) and patient-derived microglia-like cells obtained from PBMC. *CX3CR1* was significantly down-regulated in PBMC of SZ patients compared with healthy individuals [72,73]. Besides, a significant negative correlation of *CX3CR1* expression with the depression-anxiety and general psychopathology score of the Positive and Negative Syndrome Scale (PANSS), the score of the Calgary Depression Scale for Schizophrenia and the score of State and Trait Anxiety Scale of the State-Trait Anxiety Inventory was shown [72]. Ormel et al. (2020) found no significant differences between SZ and healthy individuals in *CX3CR1* gene and protein expression and in CCR2 and CCR5 protein expression in monocyte-derived microglia-like cells obtained from PMBC [74].

Thus, the available data suggest dysregulation of the expression of some chemokine receptors in SZ. The expression of chemokine receptors varies depending on the region of the brain or the analyzed cells and tissues. However, the data are conflicting and scarce. Therefore, further studies are needed to elucidate the characteristic expression patterns of chemokine receptors in SZ.

### 3.6. Animal Models Shedding Light on Chemokine Associations with SZ

Human studies cannot prove a causal relationship between chemokine dysregulation and SZ. The data described above show only the association of chemokine abnormalities with pathogenic processes in SZ. However, animal models may help elucidate the role of chemokines in the pathogenesis of SZ. We have compiled the available data in Table 6.

Schizophrenia is known to be linked to the influence of genetic and environmental factors. Exposure to viral and bacterial infections during pregnancy is a well-established risk factor for the development of schizophrenia in adult offspring [85]. Animal models of activation of maternal immunity (MAI) help to elucidate the molecular mechanisms of the influence of maternal infections in the development of schizophrenia in offspring [86]. MIA models induced by animal immunization with lipopolysaccharide (LPS, the bacterial cell-wall component) or polyinosinic polycytidylic acid (poly I:C, double stranded viral RNA analogue) revealed abnormalities in chemokine expression in different brain regions in offspring (Table 6). MIA models data indicate an important role of the CX3CL1–CX3CR1 axis in schizophrenia (extensively reviewed in [59]). Abnormal expression of CX3CR1 and CX3CL1 (fractalkine) in various animal brain regions and isolated cells was accompanied by behavioral schizophrenia-like disturbances [75,76,77,78,79,80]. Besides, prenatal maternal stress also led to deficits in CX3CL1 signaling [81]. Intracerebroventricular administration of CX3CL1 reduced the behavioral changes caused by prenatal stress [81]. Interestingly, there is evidence of decreased CX3CL1 expression in the brain of patients with schizophrenia [57,60]. Thus, the influence of environmental factors on the risk of schizophrenia may be mediated through the CX3CL1–CX3CR1 axis.

According to the neurodevelopmental hypothesis, schizophrenia is associated with abnormal brain development caused by pre- and postnatal risk factors [87]. CXCR4 and CXCL12 (CXCR4 ligand) are vital regulators of neuronal migration and brain development [88]. The first knockout experiments showed that CXCR4- and CXCL12-deficient mice showed signs of impaired neuronal migration in the cerebellar and hippocampus [89,90,91]. Later, it was shown that CXCR7 regulates accessibility CXCL12 for CXCR4 [82]. Moreover, migration of cortical interneurons was impaired in CXCR4- and CXCL12-deficient mice [82]. Dysfunction of cortical interneurons known to be associated with schizophrenia [92]. Interestingly, several human post-mortem studies have confirmed a decrease in the expression of CXCR4 (caudate nucleus) [58] and downstream signaling genes (DLPFC) [71] in the brains of patients. However, other studies failed to detect abnormalities in CXCR4 expression [71]. This can be explained by cell-specific expression patterns of CXCR4. Abnormal expression of CXCR4 can only be observed in some cells, and not in all brain tissues used for analysis.

The 22q11 deletion syndrome mouse model also revealed an association of CXCR4 with the pathogenesis of schizophrenia. Patients with 22q11 deletion syndrome suffer from cognitive impairment and anomalies in the development of the heart and lymphoid organs [93,94]. Moreover, people with this syndrome have a 12 to 80 folds higher risk of developing schizophrenia than the general population [94]. Using a mouse model, it has been shown that deletion of the 22q11.2 locus leads to impaired migration of parvalbumin-expressing interneurons caused by decreased CXCR4 expression on these cells [83]. Thus, the genetic anomalies observed in some patients with schizophrenia (22q11 deletion) [95] may be related to altered migration of parvalbumin-expressing interneurons caused by impaired CXCR4 expression.

Toritsuka et al. provided further evidence for the association of 22q11 deletion syndrome and CXCR4 dysregulation with schizophrenia. Decreased CXCR4 expression and relative CXCR4 expression per cell in the cerebral cortex of mouse embryos with a deletion of 18 orthologous genes of human 22q11 (Df1/+ mice) was found [59]. Moreover, a decrease in chemotactic response to CXCL12 (CXCR4 ligand) and disruption of dentate progenitor cell migration has been revealed [59]. DiGeorge syndrome critical region gene 8 (DGCR8) has been shown to underlie these neurodevelopmental abnormalities in Df1/+ mice. DGCR8-dependent decrease in miR-200a expression is partly responsible for decrease in CXCR4 expression and CXCR4/CXCL12 signaling deficiency [59]. Thus, DGCR8-dependent disruption of microRNA-mediated CXCR4/CXCL12 signaling may represent one of the chemokine-dependent molecular mechanisms leading to schizophrenia.

Taken together, these animal model data indicate the involvement of the CX3CL1-CX3CR1 and CXCL12-CXCR4 axes in the pathogenesis of schizophrenia. However, the evidence is still conflicting, so further research is needed to gain a deep understanding of the contribution of these signaling pathways in schizophrenia pathogenesis.

### 3.7. Chemokine Dysregulation and Neuroinflammation in SZ

Signs of neuroinflammation in SZ have been found in many studies [2,96,97]. The results of pathoanatomical studies showed an increase in the expression of many proinflammatory genes (including TNFα, IL-1β, IL-6 cytokine genes) in the brain of patients with SZ [65,98]. Besides, signs of activation and changes in the density of microglia were found in the brain of patients [99]. Moreover, infiltration by immune cells of the brain tissue was revealed [100,101,102]. In particular, CD163+ perivascular macrophages have been detected in the brain parenchyma in a subgroup of patients with “severe inflammation” [102]. Additionally, increased density of CD3+ T cells and CD20+ B cells was found in the brains of 30% of patients [100,101]. Thus, these data indicate neuroinflammation, neurovascular dysfunction, and increased BBB permeability in SZ [103].

Peripheral proinflammatory state affects the central nervous system [104]. Conversely, neuroinflammation (e.g., caused by neuroinfection) provokes a systemic proinflammatory state [105]. Interestingly, the peripheral proinflammatory state and neuroinflammation are associated neuroanatomical changes [106,107,108]. Moreover, patients with signs of peripheral inflammation suffer from more severe cognitive impairment than patients without inflammation [109,110]. This may be related to the effect of pro- and anti-inflammatory cytokines on the tryptophan–kynurenine metabolic pathway [111,112,113].

Chemokines are important players in most neuroinflammatory diseases [5,7]. Recent evidence also suggests the involvement of chemokines in the pathogenesis of SZ. The above evidence for increased/decreased blood and CSF levels of chemokines and dysregulation of chemokine and receptor expression in the brain and peripheral immune cells is summarized in Figure 2. Genetic and environmental factors (including infections) may be prerequisites for dysregulation of blood and brain chemokines in SZ (Section 3.1), which is also supported by findings in animal models (Section 3.6). Convincing evidence indicates an increase in blood CCL4, CCL2, CCL11 and CXCL8 (IL-8) in SZ [24,25,26,27,28] (Section 3.2). CCL2 are a powerful attractant of macrophages and monocytes and an activator of microglia [114]. CCL2 expression is upregulated by proinflammatory cytokines (including, IL-1, IL-4, TNF-α, IFN-γ) [115]. Therefore, an increase in proinflammatory cytokines promotes CCL2 expression. CCL4 causes adhesion and migration of lymphocytes and activation of the production of reactive oxygen species [116]. Thus, increased expression of CCL4 may contribute to the oxidative stress that has been identified in SZ [117]. CXCL8 binding to CXCR2 or CXCR1 promotes chemotaxis and neutrophil activation [61]. The observed increase in CXCL8 in the CSF of patients (Section 3.3) may contribute to the chemotaxis of immune cells into the brain parenchyma [40,41,42]. The peripheral proinflammatory state, in turn, may contribute to dysregulation of the expression of chemokine and receptor genes (Section 3.5 and Section 3.6). Mouse models have uncovered some chemokine-dependent molecular mechanisms leading to SZ (Section 3.6). In particular, the involvement of the CX3CL1-CX3CR1 and CXCL12-CXCR4 axes in the pathogenesis of SZ has been shown. Altogether, neuroinflammation and chemokine dysregulation are inextricably linked to the pathogenesis of SZ.

## 4. Conclusions

Multiple lines of evidence suggest neuroinflammation as an important component of SZ. In this systematic review, we summarize the data on the association of chemokines with neuroinflammation and the pathogenesis of SZ. The genetic association of chemokine and chemokine receptor gene polymorphisms has been shown. Besides, meta-analytical data indicate an increase in blood levels of CXCL8/IL-8, CCL2/MCP-1, CCL4/MIP-1β, CCL11/eotaxin-1 in SZ. CSF studies revealed an increase in CXCL8. Additionally, the expression of chemokines and their receptors changed in different brain regions. Anomalies in the expression of chemokines and chemokine receptors have also been found in peripheral immune cells. Mouse models have revealed molecular mechanisms associated with dysregulation of the CX3CL1-CX3CR1 and CXCL12-CXCR4 axes leading to schizophrenia. Altogether, dysregulation of chemokine expression may contribute to neuroinflammation in SZ. However, information on chemokine levels in blood/CSF or expression in the brain and other tissues of patients with SZ is still scarce and further research is needed. For example, CSF expression of chemokines is poorly understood for chemokines other than CXCL8. Expression of chemokines in the brain are also controversial. Thus, the data from this systematic review indicate the involvement of chemokines in the neurobiological processes that cause SZ.

## Figures and Tables

**Figure 1 ijms-24-02215-f001:**
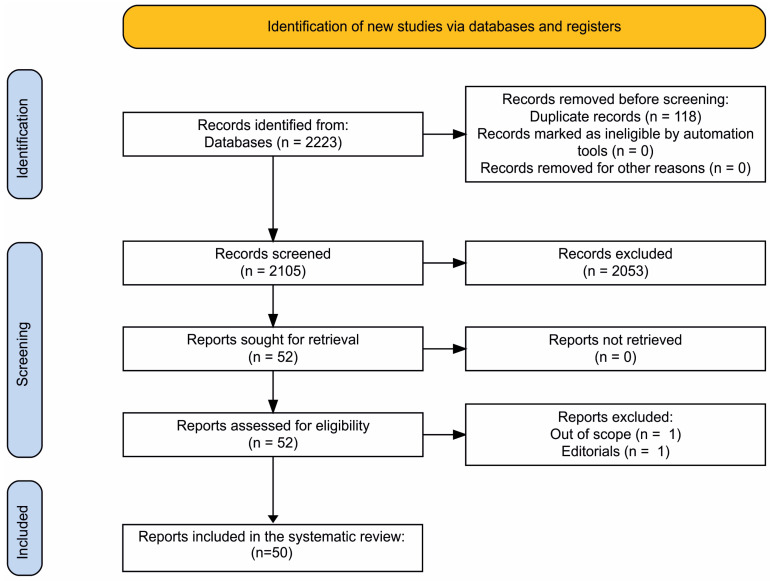
PRISMA 2020 flowchart diagram. Created using an online tool: https://estech.shinyapps.io/prisma_flowdiagram/ (accessed on 8 December 2022).

**Figure 2 ijms-24-02215-f002:**
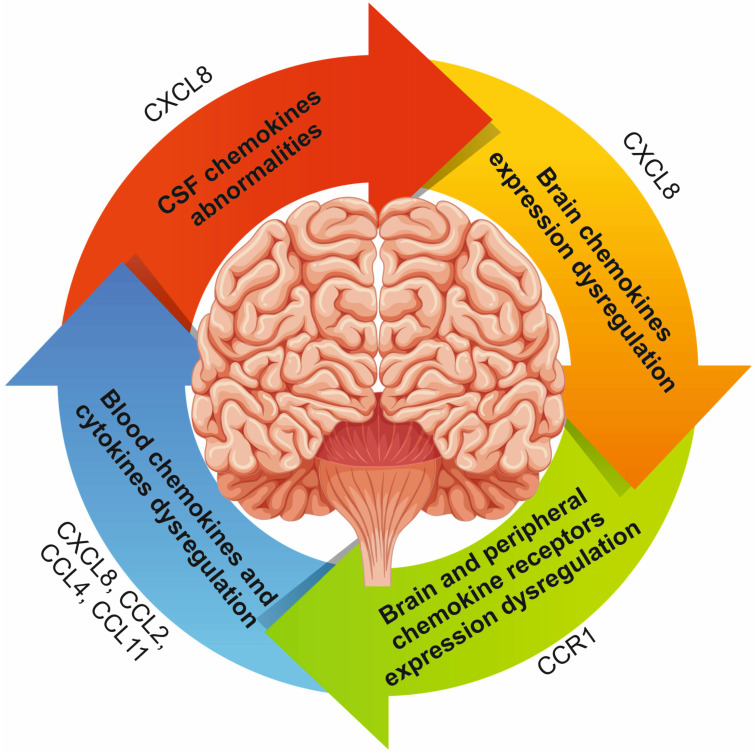
Chemokine dysregulation is associated with neuroinflammation in SZ. Blood chemokines and cytokines dysregulation contribute to CSF chemokines abnormalities. These anomalies are associated with dysregulation of the expression of chemokines and chemokine receptors in the brain and in peripheral blood lymphocytes. These processes ultimately contribute to neuroinflammation. Chemokines or receptors with altered expression in SZ are shown on the periphery of the circular arrows.

**Table 1 ijms-24-02215-t001:** Genetic association of chemokine and chemokine receptor gene polymorphisms with SZ.

Research	Analyzed Polymorphisms	Sample	Results
Saoud H et al. (2022) [11]	*CCL5*: rs2107538, rs2280788, and rs2280789; *CCR5*: rs333	200 SZ vs. 200 HS; patients and 200 controls (45.9% unSZ; 32% pSZ; 19.8% disSZ); Tunisian cohort	The rs2107538 imparted protection against SZ and unSZ; and more specifically to male sex. This SNP had an impact on patients’ symptomatology. The rs2107538, rs2280788, rs2280789 ACT genetic combination, with only one mutated allele rs2107538A, displayed reduced frequency in both SZ and unSZ. The A allele at rs2107538 could induce a protective effect by increasing RANTES production.
Saoud H et al. (2019) [12]	*CCL2* functional genetic variants of the *CCL2* (*MCP-1*) gene (MCP-1-2518A/G (rs1024611), MCP-1-362G/C (rs2857656), and MCP-1 int1del554-567 (rs3917887))	200 SZ vs. 200 HS; (45.9% unSZ; 32% pSZ; 19.8% disSZ); Tunisian cohort	The minor alleles (-2518G and Del554-567) were significantly more prevalent in HS than in SZ, whereas, for -362C minor allele, increased risk of SZ was revealed. The haplotype combination -2581G/-362G/int1del554-567 could mediate protection against SZ and the effect could result more strongly from the MCP-1 -2582G with -362G variants, whereas the effect of int1del554-567 may in part be explained by its LD with -362.
Xiong Y et al. (2014) [13]	Four SNPs of *CCL2* gene: rs4795893, rs1024611, rs4586 and rs2857657	208 SZ (two independent cohorts); Chinese cohort	This is pharmacogenetic research without HS. All genotyped SNPs were significantly associated with risperidone treatment. There may be some effect of variations in the *CCL2* gene on therapeutic efficacy of risperidone, and the associated polymorphisms may be a potential genetic marker for predicting the therapeutic effect of risperidone.
Pae CU et al. (2004) [14]	*CCL2* (MCP-1-2518A/G (rs1024611))	123 SZ vs. 114 HS; Korean cohort	Distributions of genotypes and alleles were marginally different in subjects with positive and negative symptomatology.
Mundo E et al. (2005) [15]	*CCL2* (MCP-1-2518A/G (rs1024611)	191 SZ or SAD vs. 161 HS	No significant genotypic association was found between the A-2518G variant of the *SCYA2* and the diagnosis. No differences in the age at onset of SZ were found between the three genotype groups identified. Significant genotypic association was found between the A-2518G variant of the *SCYA2* and the resistance to antipsychotic treatment, with resistant patients more frequently carrying the G allele.
Dasdemir S. et al. (2016) [16]	*CCL2*, *CCR5*, *CXCR4*, *CCR2* (MCP-1-A2518G, SDF-1-3′A, CCR5-delta32, CCR5-A55029G, CXCR4-C138T and CCR2-G190A)	140 SZ vs. 123 HS; Tunisian cohort	Frequencies of CCR5-A55029G A genotypes and CCR5-A55029G AG genotypes were found higher in SZ than the HS and even also CCR2-G190A WT: CCR5-A55029G A and CCR2-G190A A: CCR5-A55029G A haplotypes significantly associated according to Bonferroni correction. CCR5-A55029G polymorphisms and CCR2-G190A WT: CCR5-A55029G A and CCR2-G190A A: CCR5-A55029G A haplotypes might have association with SZ.
Zakharyan R et al. (2012) [17]	*CCL2* (MCP-1−2518A/G (rs1024611)	103 pSZ vs. 105 HS; Armenian cohort	The association of the −2518A/G genetic polymorphism and increased plasma levels of MCP-1 with pSZ and nominate −2518*G minor allele as a risk factor for pSZ.
Kang WS et al. (2018) [18]	*CCL11* four promoter SNPs [rs17809012 (-384T>C), rs16969415 (-426C>T), rs17735961 (-488C>A), and rs4795896 (576G>A)] in *CCL11* gene	254 SZ vs. 405 HS; Korean cohort	The genotype frequency of *CCL11* rs4795896 (-576G>A) showed significant association with SZ in a recessive model (AA vs. GG/AG, *p* < 0.0001) and in a log-additive model (AG vs. AA vs. GG, *p* < 0.0001). The allele frequency of rs4795896 also showed a significant association with SZ. The GCT, ACT, and GCC haplotypes containing rs4795896, rs17735961 and rs17809012 were significantly associated with SZ.
Ishizuka K et al. (2017) [19]	*CX3CR1* (-414G>T (rs758302878); -335G>C (rs201442030), --163G>A(rs750585901)	2283 SZ vs. 3827 HS (for GAS); 370 SZ (for TRS); Japanese cohort	Significant association between the -163G>A in *CX3CR1* gene with SZ
Kim SK et al. (2010) [20]	*CCR3* gene region (intron 1; rs9853223, rs6441948, rs13326331, and rs7652290, intron 2; rs1491962).	218 SZ vs. 377 HS; Korean cohort	The *CCR3* gene do not contribute to a genetic susceptibility to SZ
Ben Afia et al. (2020) [21]	*CXCL8* (IL-8) rs4073, rs2227306 and rs1126647	206 pSZ vs. 195 HS; Tunisian cohort	Only rs1126647 showed a significant risk for SZ.

Abbreviations: HS: healthy subjects; SZ: schizophrenia patients; unSZ: undifferentiated subtype of schizophrenia; pSZ: paranoid subtype of schizophrenia; disSZ: disorganized subtype of schizophrenia; SAD: schizoaffective disorder, depressive subtype; GAS: genetic association study; TRS: targeted resequencing study.

**Table 2 ijms-24-02215-t002:** Meta-analytic data on blood levels of chemokines in patients with SZ compared with healthy controls.

Chemokine	SZ Type	Meta-Analysis	Sample	Results
CXCL8 (IL-8)	FEP	Goldsmith et al. (2016) [25]	2 reports (49 SZ vs. 49 HS)	↑
		Frydecka et al. (2018) [24]	3 reports (99 SZ vs. 80 HS)	–
		Pillinger et al. (2019) [26]	5 reports (96 SZ vs. 145 HS)	–
		Çakici et al. (2020) [27]	6 reports (123 SZ vs. 262 HS)	↑
	ARCh	Miller et al. (2011) [28]	2 reports (46 SZ vs. 52 HS)	↑
		Goldsmith et al. (2016) [25]	2 reports (49 SZ vs. 49 HS)	↑
		Frydecka et al. (2018) [24]	12 reports (696 SZ vs. 728 HS)	↑
	FHR, CHR, or UHR	Park et al. (2019) [29]	3 reports (47 SZ vs. 97 HS)	–
		Misiak et al. (2021) [30]	5 reports (130 SZ vs. 104 HS)	–
CCL2 (MCP-1)	FEP and ARCh	Frydecka et al. (2018) [24]	11 reports (633 SZ vs. 790 HS)	↑
	FEP	Çakici et al. (2020) [27]	3 reports (57 SZ vs. 127 HS)	–
		Frydecka et al. (2018) [24]	6 reports (212 SZ vs. 265 HS)	↑
	ARCh	Frydecka et al. (2018) [24]	5 reports (421 SZ vs. 525 HS)	↑
CCL3 (MIP-1α)	FEP and ARCh	Frydecka et al. (2018) [24]	4 reports (279 SZ vs. 239 HS)	–
	ARCh	Frydecka et al. (2018) [24]	3 reports (242 SZ vs. 221 HS)	–
CCL4 (MIP-1β)	FEP and ARCh	Frydecka et al. (2018) [24]	3 reports (218 SZ vs. 150 HS)	↑
	ARCh	Frydecka et al. (2018) [24]	2 reports (181 SZ vs. 132 HS)	–
CCL11 (eotaxin 1)	FEP and ARCh	Frydecka et al. (2018) [24]	4 reports (272 SZ vs. 223 HS)	↑
	ARCh	Frydecka et al. (2018) [24]	3 reports (235 SZ vs. 186 HS)	↑
CX3CL1 (fractalkine)	FEP and ARCh	Frydecka et al. (2018) [24]	3 reports (217 SZ vs. 150 HS)	–
	ARCh	Frydecka et al. (2018) [24]	2 reports (180 SZ vs. 113 HS)	–
CXCL10 (IP-10)	FEP and ARCh	Frydecka et al. (2018) [24]	4 reports (272 SZ vs. 268 HS)	–
	ARCh	Frydecka et al. (2018) [24]	3 reports (235 SZ vs. 250 HS)	–

Abbreviations: ARCh—acute relapsed chronic patients, CHR—clinical high-risk of psychosis, FHR—familial high risk of psychosis, UHR—ultra-high-risk of psychosis, FEP—first episode psychosis, HS—healthy subjects, SZ—schizophrenia patients, ↑—statistically significantly increased level, “–”—not significant changes.

**Table 3 ijms-24-02215-t003:** Available data on the level of chemokines in the cerebrospinal fluid of patients with SZ and healthy controls.

Research	Analyzed Chemokines	Sample	Study Type	Results
Wang et al. (2018) [42]	CXCL8 (IL-8)	3 reports (112 SZ vs. 101 HS)	Meta-analysis	↑
Gallego et al. (2018) [41]	CXCL8	4 reports (105 SZ vs. 112 HS)	Meta-analysis	↑
Orlovska-Waast et al. (2019) [40]	CXCL8	3 reports (95 SZ vs. 102 HS)	Meta-analysis	↑
Nikkilä et al. (2002) [43]	CCL3 (MIP-1α)	8 SZ vs. 8 HS	Case-control	–
Malmqvist et al. (2019) [44]	CXCL10 (IP-10)	39 SZ (23 FEP) vs. 21 HS	Case-control	–

Abbreviations: FEP—first episode psychosis, HS—healthy subjects, SZ—schizophrenia patients, ↑—statistically significantly increased level, “–”—not significant changes.

**Table 4 ijms-24-02215-t004:** Chemokine expression in the brain of patients with SZ.

Research	Analyzed Chemokines	Sample	Brain Region	Method	Results
Fillman et al. (2013) [47]	CXCL8 (IL-8)	20 SZ vs. 20 HS	DLPFC (BA 46)	RT–PCR	↓
Fillman et al. (2014) [48]	CXCL8	35 SZ vs. 35 HS vs. 35 BD	Middle frontal gyrus	RT–PCR	↓
Volk et al. (2015) [49]	CXCL8	62 SZ vs. 62 HS	Prefrontal cortex (BA 9)	RT–PCR	–
Pandey et al. (2018) [50]	CXCL8	31 SZ vs. 24 HS	Prefrontal cortex (BA 9)	Western blot and RT–PCR	↓ (protein levels) – (mRNA levels)
Purves-Tyson et al. (2021) [51]	CXCL8	28 SZ vs. 29 HS	Substantia nigra	RT–PCR	–
Zhu Y. et al. (2022) [52]	CCL2, CXCL8	72 SZ vs. 69 HS	DLPFC (BA 46)	RT–PCR	*CCL2*↑ *CXCL8* ↓
Zhu et al. (2022) [53]	CXCL8	35 SZ vs. 35 HS	Ventral midbrain	RT–PCR	–
Childers et al. (2022) [54]	CXCL1, CXCL2, CXCL8, CCL2	Two RNAseq datasets used: PolyA: 183 HS vs. 168 SZ; RiboZ: 194 HS vs. 142 SZ;	DLPFC	Reanalysis of RNAseq datasets	↑ in the Type 2 SZ subgroup (proinflammatory).
Nakatani et al. (2006) [55]	CCL3	7 SZ vs. 7 HS	DLPFC (BA 46)	RT–PCR	↓
Schmitt et al. (2010) [56]	CCL2, CCL3, CCL8, CXCL8	10 SZ vs. 10 HS	Superior temporal cortex (BA 22)	Microarray	↓
Hill et al. (2021) [57]	CX3CL1	35 SZ vs. 35 HS	DLPFC	Western blot and ddPCR	↓ (protein levels) – (mRNA levels)
Weissleder et al. (2020) [58]	CXCL12	33 SZ vs. 32 BD vs. 33 HS	Subependymal zone and caudate nucleus	RT–PCR	↓
Toritsuka et al. (2013) [59]	CXCL12	18 SZ vs. 18 HS	Olfactory neuronal layer	Microarray	↓
Gandal et al. (2018) [60]	Numerous CC, CXC, C, and CX3C chemokines	159 SZ vs. 293 HS	Cerebral cortex	Meta- analysis of microarray data	*CCL3*, *CCL4*, *CXCL12*, *CXCL14*, and *CX3CL1* ↓ *CCL23* ↑

Abbreviations: BA—Brodmann area, BD—bipolar disorder, ddPCR—droplet digital polymerase chain reaction, DLPFC—dorsolateral prefrontal cortex, HS—healthy subjects, SZ—schizophrenia patients, RT–PCR—reverse transcription polymerase chain reaction, ↑—statistically significantly increased level, ↓—statistically significantly decreased level, “–”—not significant changes.

**Table 5 ijms-24-02215-t005:** Expression of chemokine receptors in the brain and other tissues of patients with SZ.

Research	Analyzed Chemokine Receptors	Sample	Tissue/Cells	Method	Results
Brain tissues					
Schmitt et al. (2010) [56]	CCR1	10 SZ vs. 10 HS	Left superior temporal cortex	Microarray	↓
Volk et al. (2015) [67]	CXCR4, CXCR7	62 SZ vs. 62 HS	PCF (BA 9)	RT–PCR	*CXCR7* ↑ *CXCR4* ↑ (trend)
Hill et al. (2021) [57]	CX3CR1	35 SZ vs. 35 HS	PFC	Western blotting and ddPCR	– (mRNA levels) protein levels were correlated with pre-synaptic protein SNAP-25 levels in SZ
Zhang et al. (2020) [68]	CX3CR1	35 SZ vs. 34 HS	Anterior cingulate cortex; DLPFC	RT–PCR	– in DLPFC. ↑ in anterior cingulate cortex of SCZ patients without suicide compared to suicide completers.
Weissleder et al. (2020) [58]	CXCR4, CXCR7	33 SZ vs. 33 HS	Subependymal zone; caudate nucleus	RT–PCR	↓
Aston et al. (2004) [69]	CX3CR1	12 SZ vs. 14 HS	Temporal cortex (BA21)	Microarray	↓
Li et al. (2016) [70]	CX3CR1	10 SZ datasets from GEO-NCBI	Hippocampus and other brain regions	Reanalysis of RNAseq datasets	↓ in hippocampus – in other brain regions
Lanz et al. (2019) [71]	CXCR4	19 SZ vs. 19 HS	DLPFC (BA 46), dorsal striatum, and hippocampus	RT–PCR	– downstream signaling genes and *CXCL12* (CXCR4 ligand) ↓ in the DLPFC
Toritsuka et al. (2013) [59]	CXCR4	18 SZ vs. 18 HS	Olfactory neuronal layer	Microarray	–
Gandal et al. (2018) [60]	Numerous CC, CXC, C, and CX3C chemokines	159 SZ vs. 293 HS	Cerebral cortex	Meta- analysis of microarray and RNAsec data	*CCR8*, *XCR1* ↑ *CX3CR1*↓
Other tissues and cells					
Bergon et al. (2015) [72]	CX3CR1	29 SZ vs. 31 HS	PBMC	RT–PCR	↓
Fries et al. (2018) [73]	CX3CR1	52 SZ vs. 20 HS	PBMC	Microarray	↓
Ormel et al. (2020) [74]	CX3CR1, CCR2, CCR5	20 SZ vs. 20 HS	Patient-derived microglia-like cells obtained from PBMC	RNAseq and mass cytometry	CX3CR1—(mRNA and protein levels) CCR2, CCR5—(protein levels)

Abbreviations: BA—Brodmann area, ddPCR—droplet digital polymerase chain reaction, DLPFC—dorsolateral prefrontal cortex, HS—healthy subjects, PBMC—peripheral blood mononuclear cells, PFC—prefrontal cortex, SZ—schizophrenia patients, RNAseq—RNA sequencing, RT–PCR—reverse transcription polymerase chain reaction, ↑—statistically significantly increased level, ↓—statistically significantly decreased level, “–”—not significant changes.

**Table 6 ijms-24-02215-t006:** Animal models data showing the association of chemokine abnormalities in the pathogenesis of SZ.

Animal Model *	Observed Chemokine-Related Abnormalities	Association with Schizophrenia	Reference
LPS- and Poly I:C-induced MIA models	Abnormal gene and protein expression of CX3CL1 and CX3CR1 in the hippocampus and frontal cortex in young and adult offspring	The mice had behavioral schizophrenia-like disturbances including PPI deficiency and an aggressive phenotype in adulthood	[75]
LPS-induced MIA model	Abnormal gene and protein expression of CX3CL1 and CX3CR1 in the hippocampus and frontal cortex in adult offspring with and without a deficit in PPI and after additional challenge with LPS	The mice had behavioral schizophrenia-like disturbances including increased exploratory activity and anxiety-like behaviors	[76]
Poly I:C-induced MIA model	Abnormal gene and protein expression of CX3CL1 and CX3CR1 in the hippocampus and frontal cortex in adult offspring with and without a deficit in PPI and after additional challenge with Poly I:C	The mice had increased exploratory activity, depressive-like episodes, diminished number of aggressive interactions	[77]
Poly I:C-induced MIA model	“–” *CX3CL1* and *CX3CR1* expression in cerebral cortex, cerebellum, and hippocampus	The mice had partly sex-dependent behavioral schizophrenia-like disturbances including deficits in PPI, increased repetitive behavior, reduced sociability, and anxiety	[78]
Poly I:C-induced MIA model	↓ *CCL2* and *CX3CR1* expression in microglial cells isolated from the hippocampus in adult offspring	Some mice had working memory impairment and deficits in social behavior and PPI	[79]
Poly I:C-induced MIA model	↑ frequency of microglial cells expressing CCR2 and CX3CR1 (trend) in adolescent female descendants ↓ CX3CR1 expression in male offspring both in adolescence and adulthood “–” CCR7 expression in adolescence and adulthood	Adult female offspring had deficits in PPI	[80]
Prenatal maternal stress rat model	Prenatal stress leads to deficits in CX3CL1 signaling Intracerebroventricular administration of CX3CL1 attenuates the behavioral changes caused by prenatal stress	Prenatal maternal stress increases risk of schizophrenia in offspring	[81]
CXCR4- and CXCL12-deficient mice	CXCR4- or CXCL12- deficiency are associated with impaired cortical interneuron migration	Cortical interneurons are impaired in schizophrenia	[82]
22q11 deletion syndrome mouse model	CXCR4-dependent impairment of parvalbumin-expressing interneuron migration ↓ CXCR4 expression on migrating interneurons	Impaired migration of parvalbumin-expressing interneurons is associated with schizophrenia	[83]
22q11 deletion syndrome mouse model	↓ CXCR4 expression in the cerebral cortex of embryos; ↓ relative expression of CXCR4 per cell ↓ chemotactic response to CXCL12 and disruption of dentate progenitor cell migration DGCR8-dependent ↓ *miR-200a* expression is partly responsible for ↓ CXCR4 expression and CXCR4/CXCL12 signaling deficiency	22q11 deletion syndrome in mice and humans is accompanied by behavioral disturbances and interneuron migration abnormalities characteristic of schizophrenia	[59]

Abbreviations: Dgcr8—DiGeorge syndrome critical region gene 8, LPS—lipopolysaccharide, MIA—maternal immune activation, PPI—prepulse inhibition, poly I:C—polyinosinic polycytidylic acid, ↑—statistically significantly increased level, ↓—statistically significantly decreased level, “–”—not significant changes. * Some MIA model data is extracted from Chamera et al. [84].

## Data Availability

All data analyzed in this study are publicly available and previously published. References to these works are included in this article. Additional information is available on request from the corresponding authors.

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
