# Peer review of "Chemokine Dysregulation and Neuroinflammation in Schizophrenia: A Systematic Review"

_ijms, 2023, doi:10.3390/ijms24032215_

Round 1

Reviewer 1 Report

The manuscript focuses on an interesting topic and the authors comprehensively summarized the issue. 

I have only one objection about how the differences in chemokine levels between the groups are described. The authors, both in the text and in the titles of the tables, consistently use the term "changes in concentration/expression in patients with schizophrenia", which suggests prospective studies and comparisons of chemokine levels over different periods of time. This term is misleading because the authors present differences between the groups (SZ vs HC) and not changes due to the appearance of the disease in given patient.

Minor remarks, the article should be corrected for punctuation (e.g. a period is missing in lines 102 and 105 after "et al")

Author Response

Dear Reviewer,

The authors deeply appreciate your thorough analysis of manuscript.

Below we answer your suggestions point by point. Your comments are in italics.

The manuscript focuses on an interesting topic and the authors comprehensively summarized the issue.

Reply: Thank you for your high appreciation of our manuscript.

I have only one objection about how the differences in chemokine levels between the groups are described. The authors, both in the text and in the titles of the tables, consistently use the term "changes in concentration/expression in patients with schizophrenia", which suggests prospective studies and comparisons of chemokine levels over different periods of time. This term is misleading because the authors present differences between the groups (SZ vs HC) and not changes due to the appearance of the disease in given patient.

Reply: We agree that the term "changes in concentration/expression" is not correct in this case. We have changed the titles of sections, tables and descriptions throughout the text. Thanks for this suggestion.

Minor remarks, the article should be corrected for punctuation (e.g. a period is missing in lines 102 and 105 after "et al")

Reply: We checked the text and found a few more typos. All of them have been corrected.

Thanks again for your valuable recommendations.

Best regards

Reviewer 2 Report

In their systematic review, the authors discussed studies that found an association between genetic polymorphisms in chemokine genes and deregulation of chemokine expression with schizophrenia (SZ).

Major Points.

The authors discussed the results of studies that were only able to identify an association between chemokine variants and pathogenic processes in schizophrenia, but not causality. Since a number of models (cellular, animal) are used to study the molecular mechanisms of SZ, there should be models to study the molecular mechanisms linking chemokine dysregulation to SZ. The authors have already mentioned such work by Toritsuka et al (2013) [59], but have not discussed the results obtained in the mouse model created in this work. The authors need to search for articles describing cellular or animal models and critically discuss their results, comparing them with association studies.

Figure 1: The reasons for excluding the 2053 entries in the second step of the analysis are not justified. What did the authors mean by the phrase "The first 50 results of each query were analyzed." How were the queries ordered? Why were only the first 50 results taken for analysis and not 100 or 150?

Minor comments.

Please abbreviate schizophrenia at the beginning of the introduction and use this abbreviation throughout the text and tables.

Table 1 Needs a complete revision.

- Please correct the SNPs designation: rs2107538-rs, 2280788-rs.

- Please also check SNPs designations throughout the text.

- 'rs2107538, rs2280788, and rs2280789 polymorphisms' and 'rs 333 polymorphism' the word 'polymorphism' is redundant as well as 'variants.

- Xiong Y et al (2014) [13] what is the sample size of healthy individuals?

- S. et al. (2016) [16], please provide the first author's name and sample sizes of healthy and sick people

- Ala55Thr is a protein variant, not a gene. Please specify the nucleotides and their coordinates of the polymorphic site.

- ...variant A-2518G of SCYA2 gene... - please italicize the name of the gene.

- Please correct the gene names in all tables and text.

- In Table 1 for all protein variants (e.g., Gly112Ala, Met138Ile) also indicate gene variants with nucleotide change coordinates.

- If the gene variants are in the coding region, also indicate whether they are silent or missense variants, and for missense variants, indicate the amino acid changes with protein coordinates.

Line 124 CCR2-V64I is not a gene polymorphism but an amino acid substitution of a protein. Please specify the corresponding nucleotide changes.

Line 301 CXCL8, CXCL1, CXCL2, and CCL2 are genes, not proteins, and should be italicized. Check to see if the genes and proteins are named correctly throughout the text.

Lines 472-475 Abbreviations should be defined the first time they appear in the abstract, keyword section, headings, main text, and figures/tables, but not at the end of the text.

Author Response

Dear Reviewer,

We thank the reviewer for the positive evaluation of our study and helpful criticisms and suggestions, following which we significantly modified our manuscript. We believe that these changes have significantly improved our manuscript and clarified our data presentation.

Below we answer your suggestions point by point. Your comments are in italics.

In their systematic review, the authors discussed studies that found an association between genetic polymorphisms in chemokine genes and deregulation of chemokine expression with schizophrenia (SZ).

Major Points.

The authors discussed the results of studies that were only able to identify an association between chemokine variants and pathogenic processes in schizophrenia, but not causality. Since a number of models (cellular, animal) are used to study the molecular mechanisms of SZ, there should be models to study the molecular mechanisms linking chemokine dysregulation to SZ. The authors have already mentioned such work by Toritsuka et al (2013) [59], but have not discussed the results obtained in the mouse model created in this work. The authors need to search for articles describing cellular or animal models and critically discuss their results, comparing them with association studies.

Reply: Thank you for this suggestion. Indeed, human studies cannot prove a causal relationship between chemokine dysregulation and SZ. But animal models studies have provided essential information about the association of chemokines with SZ. We have added a new section 3.6. “Animal models shedding light on chemokine associations with SZ” to the manuscript. We summarized the main results in Table 6 “Animal models data showing the association of chemokine abnormalities in the pathogenesis of SZ”. Your suggestion allowed us to expand the scope of the review and improve the manuscript.

Figure 1: The reasons for excluding the 2053 entries in the second step of the analysis are not justified. What did the authors mean by the phrase "The first 50 results of each query were analyzed." How were the queries ordered? Why were only the first 50 results taken for analysis and not 100 or 150?

Reply: Study selection for systematic review is typically a multi-stage process in which potentially eligible studies are first identified from screening titles and abstracts, then assessed through full text review. We excluded 2053 entries in the first step (during screening titles and abstracts) because they were out of scope. This is a routine practice used in almost all systematic reviews and meta-analyses. In each query, the results were automatically ranked by relevance (based on the presence of search queries in the title, abstract and keywords). We analyzed the first 50 results of each query to limit our efforts. Otherwise, you can analyze each query indefinitely. This approach is fully consistent with the PRISMA 2020 guidelines (please see https://www.bmj.com/content/372/bmj.n160). Limited review of results has been used in many systematic reviews and meta-analyses, for example in https://bmjopen.bmj.com/content/9/12/e031763.abstract. We limited ourselves to 50 results for each query, as there were many combinations of search queries. Also, the most relevant articles are usually found in the first 50 results of a query.

We have added new data on the association of chemokines with schizophrenia from animal models, so PRISMA 2020 flowchart diagram (Figure 1) has been modified.

Minor comments.

Please abbreviate schizophrenia at the beginning of the introduction and use this abbreviation throughout the text and tables.

Reply: We have introduced the abbreviation "SZ" for the term "schizophrenia".

Table 1 Needs a complete revision.

- Please correct the SNPs designation: rs2107538-rs, 2280788-rs.

Reply: We corrected the SNPs designation: rs2107538-rs, 2280788-rs and changed on rs2107538, rs2280788, rs2280789.

- Please also check SNPs designations throughout the text.

Reply: We checked SNPs designations throughout the text.

- 'rs2107538, rs2280788, and rs2280789 polymorphisms' and 'rs 333 polymorphism' the word 'polymorphism' is redundant as well as 'variants.

Reply: We used phrases from the original articles which we cited in the text “we aimed to investigate the impact of rs2107538, rs2280788, and rs2280789 polymorphisms in CCL5 gene,” (Saoud H et al. (2022) [11] https://pubmed.ncbi.nlm.nih.gov/35476033/); “We aimed to investigate a potential link between chemokines and schizophrenia and analyze the role of MCP-1-A2518G, SDF-1-3'A, CCR5-delta32, CCR5-A55029G, CXCR4-C138T and CCR2-V64I gene polymorphisms in the Turkish population.” (Dasdemir S. et al. (2016) [16] https://pubmed.ncbi.nlm.nih.gov/26906930/); “The MCP-1-2518A/G (rs1024611) polymorphism and blood levels of MCP-1 in patients with paranoid schizophrenia and healthy subjects were evaluated and compared” (Zakharyan R et al. (2012) [17] https://pubmed.ncbi.nlm.nih.gov/22425139/) etc.

But we deleted the word 'polymorphism' in the table.

- Xiong Y et al (2014) [13] what is the sample size of healthy individuals?

Reply: The study of Xiong Y et al (2014) [13] is pharmacogenetic research without healthy control. This is common situation for pharmacogenetic studies to compare efficacy of pharmacotherapy or side effects within only patients group. We added phrase in text.

- S. et al. (2016) [16], please provide the first author's name and sample sizes of healthy and sick people

Reply: This information presented in the table 1  (Dasdemir S. et al. (2016) [16] and 140 SZ vs. 123 HS;)

- Ala55Thr is a protein variant, not a gene. Please specify the nucleotides and their coordinates of the polymorphic site.

Reply: We used phrases from the original article which we cited in the text “We detected a statistically significant association between the variant Ala55Thr in CX3CR1 with SCZ and ASD phenotypes (odds ratio=8.3, P=0.020)” (https://pubmed.ncbi.nlm.nih.gov/28763059/).

But we changed the authors description their results according the request of reviewer and changed “variant Ala55Thr” on “-163G>A”. “Significant association between the -163G>A in CX3CR1 gene with SZ”

- ...variant A-2518G of SCYA2 gene... - please italicize the name of the gene.

Reply: We italicized the name of the gene.

- Please correct the gene names in all tables and text.

Reply: We corrected the gene names in all tables and text.

- In Table 1 for all protein variants (e.g., Gly112Ala, Met138Ile) also indicate gene variants with nucleotide change coordinates.

Reply: We used phrases from the original articles which we cited in the text, but we removed Gly112Ala, Met138Ile and changed on -335G>C and -414G>T

- If the gene variants are in the coding region, also indicate whether they are silent or missense variants, and for missense variants, indicate the amino acid changes with protein coordinates.

Reply: We indicated all missense variants in the text and the amino acid changes with protein coordinates. Only one rs4586 in CCL2 gene is silent mutation.

Line 124 CCR2-V64I is not a gene polymorphism but an amino acid substitution of a protein. Please specify the corresponding nucleotide changes.

Reply: We used phrases from the original articles which we cited in the text (“ We aimed to investigate a potential link between chemokines and schizophrenia and analyze the role of MCP-1-A2518G, SDF-1-3'A, CCR5-delta32, CCR5-A55029G, CXCR4-C138T and CCR2-V64I gene polymorphisms in the Turkish population.”https://www.tandfonline.com/doi/abs/10.3109/08039488.2016.1141981?journalCode=ipsc20)

We changed CCR2-V64I on CCR2-G190A and added information “A point mutation is known as CCR2-V64I (rs1799864), in which guanine-to-adenine substitution leads to replacing amino acid valine (V) on isoleucine (I).

Line 301 CXCL8, CXCL1, CXCL2, and CCL2 are genes, not proteins, and should be italicized. Check to see if the genes and proteins are named correctly throughout the text.

Reply: We italicize the name of the gene.

Lines 472-475 Abbreviations should be defined the first time they appear in the abstract, keyword section, headings, main text, and figures/tables, but not at the end of the text.

Reply: We have removed these abbreviations. They were defined the first time they appear.

Thank you again for your thorough analysis of our manuscript.

Best regards

Round 2

Reviewer 2 Report

The authors have significantly improved the manuscript as suggested by the reviewers, and I can recommend it for publication in the Journal.